# OpenReview forum: "Bridging Pairwise and Pointwise GRMs: Preference-Aware Reward Mechanism with Dynamic Rubric Adaptation"
_ICLR.cc/2026/Conference — ICLR 2026 Conference Withdrawn Submission_

### Official Review · Reviewer_Y2SM · 2025-10-29

**Soundness:** 2
**Presentation:** 3
**Contribution:** 2
**Rating:** 4
**Confidence:** 4

**Summary:**

The paper introduces PaTaRM, a Preference-Aware Task-Adaptive Reward Model that combines a preference-derived point-wise reward mechanism with dynamic rubric adaptation. PaTaRM converts relative preference data into robust point-wise training signals without requiring explicit point-wise labels, and uses a task-adaptive rubric to generate evaluation criteria that capture both global task consistency and instance-specific fine-grained reasoning. Experiments across multiple models and benchmarks indicate effectiveness.

**Strengths:**

1. Conducts experiments across different base models and benchmarks, suggesting cross-model applicability.

2. Leverages relative preferences to construct point-wise signals.

3. Employs dynamic, task-adaptive rubrics that can provide interpretable, fine-grained evaluation criteria aligned to each instance.

4. Framework appears modular and compatible with existing RLHF pipelines.

**Weaknesses:**

1. Generative reward modeling increases training and inference latency, especially during policy optimization.

2. The requirement of n generative rollouts per prompt further amplifies computational costs and may limit scalability.

3. Reward quality is limited by the evaluation LLM’s capabilities; any biases or inaccuracies in the evaluator propagate to the reward.

4. Dynamic rubric adaptation is sensitive to prompt design and may introduce prompt-induced variance or rubric drift.

**Questions:**

1.	How do you formally define the gap between pair-wise and point-wise reward models in your setting?

2.	In Figure 1, beyond illustrating differences, can you quantify the discrepancy between pair-wise and point-wise signals?

3.	What is the end-to-end compute breakdown (per prompt rollouts, evaluator LLM invocations, training updates), and how does cost scale with n rollouts, model size, and sequence length?

4.	What are the wall-clock time relative to a standard non-generative reward model?

5.	How sensitive is performance to decoding settings for the generative evaluator?

6.	Does dynamic rubric adaptation mitigate or exacerbate known reward hacking patterns compared to static rubrics or standard RMs?

7.	How sensitive are results to the evaluator LLM choice and to the rubric prompt template? Do small prompt edits materially change outcomes?

---

> ### Author Response · Authors · 2025-11-21
>
> ## **Title: Thank you for responses. We look forward to hearing back from you.**
>
> Dear Reviewer Y2SM:
>
> Thank you for your responses and valuable questions. Below we provide our point-by-point responses.
>
> **Q1: How do you formally define the gap between pair-wise and point-wise reward models in your setting?**
>
> A: Thank you for your question. To clarify the differences between the pointwise reward model and the pairwise reward model, we have summarized the key points in the following table:
> | Comparison Point | Pointwise Reward Model | Pairwise Reward Model |
> |---|---|---|
> | **Core Idea** | Assigns an absolute score to a single input. | Decides which input of a pair is preferred. |
> | **Formal Definition** | $\text{Loss}_{\text{pt}}=\sum_i L(f(x_i),r_i)$ | $L(P_\theta(x_i \succ x_j), \mathbb{I}\{r_i>r_j\})$ |
> | **Input-Output Format** | Input: single sample $x_i$  <br> Output: scalar score $f(x_i)$ | Input: ordered pair $(x_i,x_j)$  <br> Output: probability $P_\theta(x_i\succ x_j)\in[0,1]$ that $x_i$ is better than $x_j$ |
> | **Training Objective** | Minimize the distance between $f(x_i)$ and the ground-truth absolute score $r_i$. | Maximize the likelihood that the predicted preference matches the human ranking $\mathbb{I}[r_i>r_j]$. |
> | **Intuition** | Independent regression to a cardinal score. | Binary classification on the relative order. |
>
>
>
>
> **Q2: In Figure 1, beyond illustrating differences, can you quantify the discrepancy between pair-wise and point-wise signals？**
>
> A: Thank you for your question. In addition to illustrating the differences, we can quantify the gap between pairwise and pointwise signals through the following points:
>
> 1. **Quantifying computational complexity**:
>    - In GRPO scenarios, assuming there are $n$ rollouts to be scored, pointwise models require $n$ forward passes with $O(n)$ complexity. Pairwise models require $O(n\log n)$ comparisons even with optimal sorting algorithms. For example, with $n=16$, pointwise models require 16 inferences while pairwise models require approximately 64 comparisons—a 4× overhead that grows with $n$.
>
> 2. **Quantifying values in policy integration**:
>    - The output of pairwise reward models is a relative score difference, which cannot be directly used for value function estimation in policies. In contrast, the output of pointwise reward models is an absolute score, which can be directly used for value function estimation. Therefore, pairwise models have limitations when integrating into policy value function estimation.
>
> 3. **Bridging the gap with our method**:
>    - Our method combines the strengths of both approaches: we train using high-quality pairwise data, transforming pairwise signals into pointwise supervision through the PAR mechanism, then deploy as a pointwise model with $O(n)$ inference complexity. This achieves the data quality advantages of pairwise training while maintaining the computational efficiency and LLM reasoning capabilities of pointwise evaluation.
>
> **Q3: What is the end-to-end compute breakdown (per prompt rollouts, evaluator LLM invocations, training updates), and how does cost scale with n rollouts, model size, and sequence length?**
>
> A: Thank you for your question.
> - In our downstream experiments, we performed 4 rollouts per prompt, evaluated the LLM at step 128, with a global batch size of 256, resulting in a total of 131,072 evaluations, and conducted 128 training updates, which is the same as the number of steps.
> - We tested the time cost on 16 A-100 GPUs in our PaTaRM training, as shown in the table below. We will add this information to Appendix D.
>
> | Model  | Parameter Size | Sequence Length | Rollout per Prompt | Training Time per Step (s) | Total Training Duration (h) |
> |--------|----------------|-----------------|---------------------|----------------------------|-----------------------------|
> | Qwen3  | 8B             | 4k              | 8                   | 246                        | 8.75                        |
> | Qwen3  | 8B             | 4k              | 16                  | 486                        | 17.28                       |
> | Qwen3  | 8B             | 1k              | 16                  | 311                        | 11.05                       |
> | Qwen3  | 8B             | 2k              | 16                  | 415                        | 14.11                       |
> | Qwen3  | 8B             | 4k              | 4                   | 125                        | 4.44                        |
> | Qwen3  | 14B            | 4k              | 4                   | 277                        | 9.85                        |
>
> Regarding how the cost changes with rollout numbers, model size, and sequence length, our experiments have shown the following conclusions:
>
> - Increasing the number of rollouts per prompt significantly increases the total training duration.
> - Larger models require more time per training step.
> - Longer sequence lengths also increase the training time per step.

---

> > ### Comment · Reviewer_Y2SM · 2025-11-26
> >
> > Thanks for the reply. I have a follow-up question: Why do pairwise models require O(n log n) comparisons at inference? Although the BT model is trained on preference pairs, it induces a latent score for each item. Once we compute these scores, we can rank n items with a single forward pass per item, which is O(n), rather than O(n log n).

---

> > > ### Author Response · Authors · 2025-11-26
> > >
> > > ### Response to ReviewerY2SM
> > >
> > > Thank you for your follow-up question. We apologize for any confusion caused. Your question has highlighted the need for us to clarify our model taxonomy more explicitly.
> > >
> > > It seems there might be a misunderstanding regarding the terminology we used. In our paper, we categorize reward models into two main classes:
> > >
> > > 1. **BT Scalar Models (Bradley-Terry scalar models)** - These models output a scalar score for each item. As you correctly pointed out, they can rank $ n $ items with $ O(n) $ forward passes.
> > >
> > > 2. **Generative Reward Models (GRMs)** - These are further divided into:
> > >    - **Pairwise GRMs**: Take two responses (A and B) as input and output a judgment of "which is better".
> > >    - **Pointwise GRMs**: Generate evaluations for a single response.
> > >
> > > When we mentioned **$O(n log n)$ complexity**, we were specifically referring to **Pairwise GRMs**. Since these models require response pairs (response A, response B) as input, ranking $ n $ items requires $ O(n \log n) $ comparisons (e.g., using comparison-based sorting algorithms), with each comparison requiring one model forward pass.  In contrast, BT scalar models and Pointwise GRMs indeed only require $ O(n) $ forward passes.
> > >
> > > To clarify this distinction, we provide the following comparison table:
> > >
> > > | Model Type        | Training Data Format | Inference Input Format | Inference Output Format | Ranking Complexity |
> > > |-------------------|----------------------|------------------------|-------------------------|--------------------|
> > > | **BT Scalar Model** | Preference pairs: (A,B,label)  \label ∈ {A>B, B>A} | Single response | Scalar score  \(e.g. reward ∈ ℝ\) | **O(n)**  \(n forward passes\) |
> > > | **Pairwise GRM** | Preference pairs: (A,B,label)  \label ∈ {A>B, B>A} | Response pair (A,B) | Comparative judgment  \(e.g. "Given the two response, A is better"\) | **O(n log n)**  \(sorting comparisons\) |
> > > | **Pointwise GRM** | Single responses + ratings (resp, score) | Single response | Eval text | **O(n)**  \(n forward passes\) |
> > > | **PaTaRM** | Preference pairs → converted to pointwise | Single response | Eval text | **O(n)**  \(n forward passes\) |
> > > 1. **BT Scalar Model**: Although trained on pairs, it learns a latent score for each item and can score independently at inference.
> > > 2. **Pairwise GRM**: Both training and inference are based on pairs; ranking requires multiple pairwise comparisons.
> > > 3. **Pointwise GRM**: Both training and inference operate on single responses, generating evaluation text.
> > > 4. **PaTaRM (Our Contribution)**:
> > >   - Compared to BT scalar models that also leverage pairwise data, PaTaRM provides **interpretable reasoning** rather than just outputting a score, which operates as a black-box.
> > >    - Compared to the training methods of previous pointwise GRMs, our innovative **PAR mechanism** allows PaTaRM to be trained on readily available pairwise preference data **without requiring explicit rating labels**, which are harder to obtain and can introduce bias.
> > >    - Compared to Pairwise GRMs, PaTaRM has $ O(n) $ inference complexity, making it more suitable for integration into downstream RLHF experiments.

---

> ### Author Response · Authors · 2025-11-21
>
> **Q4: What are the wall-clock time relative to a standard non-generative reward model?**
>
> A: Thank you for your question. We conducted downstream experiments(seq_len=4k，rollout=16) to compare the wall-clock time between BT model and our PaTaRM model. The results are summarized in the table below:
> PaTaRM incurs approximately 25-39% additional training time per step compared to BT models due to the generative nature of producing detailed evaluation reasoning. However, this additional cost is justified by PaTaRM's better interpretability, and stronger generalization to OOD tasks.
>
> | Model | RM Type| Time per Step (s) |
> |-|-|-|
> | Qwen3-8B     | BT      | 987  |
> | Qwen3-8B     | PataRM | 1230  |
> | Qwen3-14B    | BT  | 1149  |
> | Qwen3-14B    | PataRM | 1599   |
>
> **Q5: How sensitive is performance to decoding settings for the generative evaluator?**
>
> A:Thank you for this question.
> - We conducted comprehensive experiments to evaluate the sensitivity of our model to decoding settings, as shown in the tables below.
> - The experimental results demonstrate that while decoding settings do influence performance, the sensitivity is relatively modest.
> - Based on the experimental results, we suggest the following decoding parameter settings for optimal performance:
>     - **Qwen3-8B**: Temperature: 0.6, Top-p: 1, Top-k: -1
>     - **Qwen3-14B**: Temperature: 0.8, Top-p: 1, Top-k: -1
>
> **Temperature Sensitivity (top_k=-1, top_p=1):**
>
> | Temperature | 0 | 0.2 | 0.4 | 0.6  | 0.8 | 1.0  |
> |-|-|-|-|-|-|-|
> | Qwen3-8B    | 83.7 | 83.4 | 83.8 | 84.3 | 82.7 | 83.1 |
> | Qwen3-14B   | 85.7 | 85.2 | 85.3 | 85.9 | 86.4 | 86.0 |
>
> **Top-p Sensitivity:**
>
> | Top-p |0.6|0.7|0.8|0.9|1.0|
> |-|-|-|-|-|-|
> | Qwen3-8B (temp=0.6, top_k=-1) | 82.5 | 84.5 | 83.1 | 83.3 | 84.3 |
> | Qwen3-14B (temp=0.8, top_k=-1) | 84.1 | 85.6 | 85.2 | 85.8 | 86.4 |
>
> **Top-k Sensitivity:**
>
> | Top-k|1|5|10| 20 | -1|
> |-|-|-|-|-|-|
> | Qwen3-8B (temp=0.6, top_p=1) | 83.4 | 83.2 | 83.4 | 83.8 | 84.3 |
> | Qwen3-14B (temp=0.8, top_p=1) | 85.4 | 86.0 | 86.4 | 86.2 | 86.4 |
>
> **Q6: Does dynamic rubric adaptation mitigate or exacerbate known reward hacking patterns compared to static rubrics or standard RMs？**
>
> A: Thank you for this question. We believe dynamic rubric adaptation mitigates reward hacking compared to static rubrics and standard reward models.
>
> - **Limitations of static rubrics and standard BT models:** Predefined static rubrics force reward models to focus narrowly on specified criteria, causing them to overlook obvious errors not explicitly mentioned [1]. This rigid focus makes models vulnerable to overfitting. Additionally, Bradley-Terry (BT) models that output only scalar scores lack interpretability.
>
> - **Advantages of dynamic adaptation:** Our dynamic rubric mechanism generates task-specific evaluation dimensions contextually, capturing both predefined quality aspects and emergent issues not covered by static templates.
>
> - **Experimental validation:** In instruction-following tasks, we compared Checklist DPO (static rubrics), Skywork (scalar BT model), and PaTaRM (dynamic rubrics). PaTaRM outperforms both approaches when integrated into RL. While reward hacking cannot be explicitly quantified, this superior downstream performance provides strong evidence that dynamic adaptation effectively mitigates the phenomenon.
>
> [1] Li, W., Wang, X., Yuan, S., Xu, R., Chen, J., Dong, Q., Xiao, Y., & Yang, D.  Curse of Knowledge: When Complex Evaluation Context Benefits yet Biases LLM Judges.EMNLP 2025 Findings.
>
> **Q7: How sensitive are results to the evaluator LLM choice and to the rubric prompt template? Do small prompt edits materially change outcomes?**
> A: Thank you for your question. We conducted experiments to evaluate sensitivity to both evaluator LLM choice and prompt template variations.
>
> - **Prompt template sensitivity:** To test robustness to minor prompt modifications, we systematically replaced key terms in our prompt template: all instances of "generate" were changed to "produce" and all instances of "eval" were replaced with "evaluate". The results is shown in table below, indicating that small prompt edits do not materially change outcomes.
>
> **RewardBench Results:**
>
> | Model               | Chat | Chat Hard | Reasoning | Safety | Overall |
> |-|-|-|-|-|-|
> | PaTaRM-Qwen3-8B     | 91.0 | 71.5   | 87.9  | 86.3   | 84.2  |
> | PaTaRM-Qwen3-8B(edit)  | 90.5 | 69.3 | 89.5  | 85.5   | 83.7    |
> | PaTaRM-Qwen3-14B    | 94.0 | 73.9 | 91.7   | 85.6   | 86.3  |
> | PaTaRM-Qwen3-14B(edit)     | 91.9 | 74.3  | 91.8| 85.9   | 86.0   |
>
> - **Evaluator LLM sensitivity:** As shown in our cross-model experiments, we validated PaTaRM across different model families including Qwen3 and Llama. On rewardbench, we achieved at least a 5% relative improvement, and on RMBenchs, we achieved at least a 4% relative improvement. These consistent improvements demonstrate that our method generalizes well across different LLM architectures and is not overly sensitive to the specific evaluator choice.

---

> ### Author Response · Authors · 2025-11-26
> **Gentle reminder of the author-reviewer discussion deadline**
>
> Dear Reviewer Y2SM:
>
> Thank you for your thoughtful and insightful comments! We sincerely appreciate your valuable feedback and suggestions. We are eager to engage in further discussions with you! In our previous responses, we have addressed all of the points you raised, supplementing the relevant experiments and clarifying any ambiguities. We believe that these revisions strengthen the contributions of our paper and look forward to your thoughts.
>
> As the discussion period deadline approaches, if you have any additional questions or concerns about the paper, we would be delighted to continue the conversation with you! We sincerely hope that our responses have effectively addressed your concerns and may encourage a more favorable reconsideration of our paper.
>
> Best regards

---

### Official Review · Reviewer_YRyH · 2025-11-01

**Soundness:** 2
**Presentation:** 2
**Contribution:** 1
**Rating:** 2
**Confidence:** 4

**Summary:**

This paper introduces the Preference-aware Task-adaptive Reward Model (PaTaRM), a framework that introduces (1) a Preference-Aware Reward (PAR) mechanism that efficiently converts relative preference signals from widely available pairwise data into robust pointwise training signals, eliminating the need for explicit pointwise labels, and (2) a dynamic rubric adaptation system where the model generates its own flexible, instance-specific evaluation criteria for fine-grained reasoning. Qwen3-8B and Qwen3-14B are optimized via supervised fine-tuning and then reinforcement learning, and experiments show that the reward models have improved performance on reward model benchmarks and provide more effective reward signals for downstream RLHF policy alignment.

**Strengths:**

**S1.** The paper directly addresses the inflexibility of rubric-based GRMs that rely on rubric generation from external models.

**S2.** The models perform well on the downstream reward modeling benchmark, and the reward signals lead to policy improvements on benchmarks like IFEval and InFoBench.

**Weaknesses:**

**W1.**  The given formulation actually does not explicitly enforce global transitive consistency across preference chains, which is my concern, especially when you are trying to convert somewhat less information (pairwise) into richer information (pointwise). In practice, this means the model may learn locally coherent but globally inconsistent relationships (e.g., a > b, b > c, yet a<c), since it optimizes over independent pairwise comparisons. I believe this could lead to unstable global rankings and reduce the interpretability of the learned reward function.

**W2.** Since PaTaRM generates the instance-specific rubrics it uses for evaluation, there is a potential for reward self-optimization, where the model implicitly learns to produce rubrics that are easier for it to satisfy rather than faithfully reflecting nuanced user preferences. Without strong constraints or external validation, this adaptive rubric generation could risk a subtle form of “reward hacking” and the authors seem not to address this.

**W3.** It seems that the performance of PaTaRM is still behind compared to existing reward models, in particular R3 (see Q1: it seems like the reported numbers are mismatched with the original paper). However, R3 only utilizes SFT (possibly with high quality) without any further RL training, so I wonder if you would consider Q2.

**Questions:**

**Q1.** Is there any result for RM-Bench pairwise and comparison with the other reward models? In addition, there is a mismatch between the reported number for R3's performance on RewardBench. However, for all other numbers, I think they should be good.

**Q2.** Since the SFT seems to reduce performance, I wonder if it is because the SFT dataset is not great in terms of quality, especially since Qwen-2.5-72B-Instruct is being used. It is possible that even Qwen3-8B or Qwen3-14B already performs better than Qwen-2.5-72B-Instruct. Perhaps, you could either first distill from better models for better SFT data quality or skip the SFT training and allocate all data using RL.

**Q3.** Is there any statistical significance reported?

---

> ### Author Response · Authors · 2025-11-21
>
> Dear Reviewer YRyH:
>
> Thank you for your responses and valuable questions. Below we provide our point-by-point responses.
>
> **W1: Global Transitive Consistency Concerns.**
>
> A: Thank you for this important concern. We address it from theoretical and empirical perspectives:
>
> **1. Theoretical guarantee:** Our method implicitly ensures global transitivity through a BT-analogous reward design. For responses $a, b, c$ with ground truth $a > b > c$, we perform $K$ rollouts and compute average scores: $\bar{s}(x) = \frac{1}{K}\sum_i s_i(x)$.
>
> Our optimization objective enlarges score differences: for $(a > b)$, it increases $\bar{s}(a)$ and decreases $\bar{s}(b)$; similarly for $(b > c)$. By the transitivity of real numbers, joint optimization inevitably yields $\bar{s}(a) > \bar{s}(b) > \bar{s}(c)$, mapping pairwise comparisons onto a globally consistent reward scale.
>
> **2. Empirical validation:** During training, our model observes only 2 responses per query. During inference for policy optimization, it evaluates and ranks 16 rollouts per query—an 8× scale-up. If global inconsistency existed, this would produce chaotic signals. Instead, the model stably outputs effective rewards and successfully guides policy updates, validating its global consistency even at larger scales.
>
> **W2:Reward Self-Optimization and Reward Hacking Concerns**
>
> A:Thank you for raising this concern. We have indeed considered and explicitly addressed this potential risk, which is one of the core motivations behind our design
>
> **1. Design-Level Constraint Mechanism: Dynamic Rubric Adaptation：**
> To prevent reward self-optimization and potential reward hacking, we propose the Dynamic Rubric Adaptation mechanism, the Primary Rubric provides a stable and consistent evaluation baseline that is not influenced by the model's adaptive generation. In our prompts, we explicitly require the model to assign higher weight to the Primary Rubric.
> This design uses human-defined global constraints as an "anchor point," ensures that evaluations consistently focus on core quality dimensions.
>
> **2. Empirical Validation: Multi-Dimensional Performance：**
> Our model achieves consistent improvements over baseline models on standard benchmarks, demonstrating that the generated rubrics can effectively distinguish responses.
> More critically, our reward model exhibits strong generalization performance on out-of-distribution (OOD) tasks. It successfully guides policy model improvements on task types unseen during training This cross-task generalization ability serves as strong evidence against reward hacking.
>
> Through these design choices and validations, we believe PaTaRM effectively addresses the risks you raised and maintains reliability and stability across a wide range of application scenarios.
>
> **W3:It seems that the performance of PaTaRM is still behind compared to existing reward models, in particular R3.**
>
> A:
> We thank the reviewer for pointing out the performance gap between PaTaRM (Table 1) and R3 (Table 3).
>
> 1.We believe this comparison requires methodological clarification: PaTaRM employs a **pointwise scoring** paradigm that independently evaluates individual responses, which better aligns with practical RLHF deployment scenarios. However, R3 and similar models use **pairwise comparison**, judging relative quality between response pairs—a cognitively simpler task that typically yields higher benchmark scores.
>
> 2.Consequently, **direct comparison of leaderboard scores between pointwise and pairwise methods may not be entirely equitable**, particularly when data volume, training paradigms, and evaluation methodologies differ. To address this concern, we will include Qwen3-BT (Bradley-Terry) experiments in the revised version, conducted under matched data volumes, to validate our method within a comparable framework.
>
>  3.**This is precisely why we present results in two separate tables in our paper**—Table 1 shows results for pointwise scoring methods, while Table 3 presents pairwise comparison methods. In essence, PaTaRM and R3 represent two distinct output paradigms, each with its own applicable scenarios and advantages.

---

> ### Author Response · Authors · 2025-11-21
>
> **Q1: Is there any result for RM-Bench pairwise and comparison with the other reward models? In addition, there is a mismatch between the reported number for R3's performance on RewardBench.**
>
> A: Thank you for your questions.
>
> **1. Clarification on R3 data consistency:** We have verified that the R3 performance numbers cited in our paper are consistent with those reported in the original publication. The R3 paper presents results across multiple SFT configurations; we specifically reference their **R3-QWEN3-8B-14K** and **R3-QWEN3-14B-14K** configurations for fair comparison.
>
> **2. RM-Bench pairwise results:**
>
> - We provide the RM-Bench pairwise and comparison with the other reward models as shown in table below. While R3 achieves strong results on RM-Bench, it is not an appropriate baseline for our work due to fundamental differences: R3 employs pure supervised fine-tuning (SFT) and utilizes different training data.
> - We would note that most generative reward models (GRMs) **achieve high benchmark scores through pairwise comparison but often lack rigorous validation in downstream RLHF tasks**. Our work emphasizes **pointwise evaluation**, which better reflects real-world deployment requirements and has been thoroughly validated through policy optimization experiments. This table **is not our main results**, which serves as an anlysis experiment to prove the dynamic rubric adaptation is useful.
>
> | Model | Easy | Medium | Hard | Overall |
> |-|-|-|-|-|
> | Qwen3-8B (ours) | 87.2 | 81.7 | 74.4 | 81.1 |
> | Qwen3-14B (ours) | 87.8 | 85.5 | 76.7 | 83.3 |
> | RM-R1 Qwen7B distill-r1 | 75.9 | 73.1 | 68.1 | 72.4 |
> | RM-R1 Qwen14B distill-r1 | 86.2 | 83.6 | 74.4 | 81.5 |
> | R3-QWEN3-8B-14K | 89.4 | 84.5 | 85.3 | 82.1 |
> | R3-QWEN3-14B-14K  | 91.2 | 86.3 | 87.1 | 83.8|
>
> **Q2: Perhaps, you could either first distill from better models for better SFT data quality or skip the SFT training and allocate all data using RL.**
>
> A: Thank you for this valuable suggestion. We clarify the SFT phase design and address your concerns below:
>
> **1. SFT phase design objectives:** In the SFT+RL paradigm, the SFT phase aims to teach basic instruction-following formats and conversational norms, not improve capabilities. We intentionally use 1 epoch, larger batch sizes, and lower learning rates to avoid over-altering the pretrained knowledge distribution. Minor performance fluctuations during SFT are expected, with real improvements coming from the RL phase.**Note that the reported untrained baseline uses a different prompt from our SFT model, which may affect direct comparison.**
>
> **2. Ablation study: skipping SFT:** We conducted RL-only experiments on Qwen3-8B (Table 4). Without SFT, the model focuses excessively on learning format compliance rather than optimizing task performance, easily falling into local reward optima with superficial improvements. Therefore, SFT serves as a necessary "format alignment" step.
>
> **3. Clarification on data quality:** Regarding "distilling from better models to improve SFT data quality," we clarify that SFT data optimization is not our core contribution. Our focus is the PaTaRM method, which transforms pairwise preference data into pointwise reward signals, reducing dependence on high-quality pointwise annotations. Notably, your suggestion highlights our method's robustness: even with non-optimal SFT data, PaTaRM achieves significant benchmark improvements, indicating substantial room for further gains with higher-quality data.
>
> **Q3: Is there any statistical significance reported?**
>
> A: Thank you for this important question. We conducted four inference runs using temperature=0.7, top_k=-1, and top_p=1, and report the results with standard deviations below:
>
> |   RMBench          | Easy          | Medium        | Hard          | Overall       |
> |----------------------|---------------|---------------|---------------|---------------|
> | PaTaRM-Qwen3-8B      | 83.7 ± 0.45   | 75.2 ± 0.55   | 64.6 ± 0.25   | 74.5 ± 0.32   |
> | PaTaRM-Qwen3-14B     | 86.0 ± 0.22   | 76.9 ± 0.01   | 65.4 ± 0.24   | 76.1 ± 0.15   |
>
> | RewardBench          | Chat          | Chat Hard     | Reasoning     | Safety        | Overall       |
> |----------------------|---------------|---------------|---------------|---------------|---------------|
> | PaTaRM-Qwen3-8B      | 91.0 ± 0.53   | 71.5 ± 1.17   | 87.9 ± 0.66   | 86.3 ± 0.70   | 84.2 ± 0.31   |
> | PaTaRM-Qwen3-14B     | 94.0 ± 0.95   | 73.9 ± 0.41   | 91.7 ± 0.54   | 85.6 ± 0.38   | 86.3 ± 0.09   |
>
> The relatively small standard deviations across multiple runs demonstrate the stability and reliability of our method.

---

> ### Author Response · Authors · 2025-11-26
> **Gentle reminder of the author-reviewer discussion deadline**
>
> Dear Reviewer YRyH:
>
> Thank you for your thoughtful and insightful comments! We sincerely appreciate your valuable feedback and suggestions. We are eager to engage in further discussions with you! In our previous responses, we have addressed all of the points you raised, supplementing the relevant experiments and clarifying any ambiguities. We believe that these revisions strengthen the contributions of our paper and look forward to your thoughts.
>
> As the discussion period deadline approaches, if you have any additional questions or concerns about the paper, we would be delighted to continue the conversation with you! We sincerely hope that our responses have effectively addressed your concerns and may encourage a more favorable reconsideration of our paper.
>
> Best regards

---

### Official Review · Reviewer_6o6v · 2025-11-01

**Soundness:** 3
**Presentation:** 3
**Contribution:** 2
**Rating:** 4
**Confidence:** 3

**Summary:**

The paper introduces PaTaRM (Preference-aware Task-adaptive Reward Model), a reward-modeling framework for RLHF that turns cheap pairwise preference data into rich, pointwise-like supervision. The method involves generating multiple judgment rollouts for the chosen and rejected responses, scores them under adaptive rubrics, and positively rewards rollouts that are consistent with the human preference, effectively extracting pointwise signals without explicit pointwise labels. Experiments show that this unified setup improves over standard pairwise and pointwise generative reward models on RewardBench and RMBench and also yields better downstream RLHF rewards, supporting the claim that PaTaRM meaningfully bridges pairwise and pointwise GRMs while keeping annotation costs low.

**Strengths:**

* The paper tackles a very clear gap between efficient pairwise supervision and practically useful pointwise reward models. The proposed PaTaRM combines a preference-aware reward with dynamic rubric adaptation, allowing the model to derive rich, instance-specific supervision from ordinary pairwise data, thereby improving sample efficiency.

* Empirical results on RewardBench and RMBench with two Qwen3 sizes support the claim.
* The paper is clearly-written and well formatted.

**Weaknesses:**

* The idea of deriving pointwise-like signals from pairwise labels are not new. In standard RLHF using PPO for example, the three stages were 1) SFT training, 2) Reward model training, and 3) RL (e.g., PPO) using the reward model trained in step 2. In this framework, the reward model training in stage 2 relied on a pairwise training where we train the model to assign a higher score to the chosen. Then, with this trained reward model, we use it as a pointwise reward during stage 3. Apart from having multiple rollout during the reward model training, could the authors clarify on the difference?

* The authors mention point-wise GRM could propagate bias. However, the current framework that the authors propose could also suffer from the pairwise data bias. If the pairwise data is noisy or biased, the whole reward inherits that bias and maybe amplifies it.

* As an extension of previous comment, it would be nice to see how the method compares when the labeled pairwise data is noisy (maybe with injected noise such as randomly flipped labels), and also with respect to the size of the label data.

* It would be nice to see the performance on more reasoning heavy downstream tasks such as math and coding.

**Questions:**

see above

---

> ### Author Response · Authors · 2025-11-23
>
> ## **Title: Thank you for responses. We look forward to hearing back from you.**
>
> Dear Reviewer 6o6v:
>
> Thank you for your responses and valuable questions. Below we provide our point-by-point responses.
>
> **Q1: Could the authors clarify on the difference between BT and PaTaRM?**
>
> A: Thank you for this important question. We clarify the fundamental differences between PaTaRM and the Bradley-Terry (BT) model from three key perspectives:
>
> **1. Training objective:**
>
> **BT Model:**
> BT adopts a scalar discriminative paradigm that directly optimizes score differences. Given a preference pair where $x_i$ is preferred over $x_j$ (denoted as $x_i \succ x_j$), the probability that the model agrees with this preference is:
>
> $P_\theta(i \succ j) = \sigma(r_\theta(x_i) - r_\theta(x_j)), \quad \sigma(z) = \frac{1}{1 + e^{-z}}$
>
> The negative log-likelihood (NLL) loss over $N$ preference pairs is:
>
> $L_{BT} = -\frac{1}{N} \sum_{(i \succ j)} \log \sigma(r_\theta(x_i) - r_\theta(x_j))$
>
> **PaTaRM:**
> PaTaRM adopts a generative reasoning paradigm with preference-aware optimization. Using the GRPO formulation, given a prompt $x$ and a group of $K$ sampled outputs $\{y_1, ..., y_K\}$ from $\pi_\theta$, the loss function is defined as:
>
> $L_{\text{PaTaRM}} = -\mathbb{E}\_{x \sim \mathcal{D}, \{y_k\}\_{k=1}^K \sim \pi_\theta(\cdot|x)} \left[ \frac{1}{K} \sum_{k=1}^K \hat{A}\_k \log \pi_\theta(y_k|x) \right]$
>
> where the advantage function is computed as:
>
> $\hat{A}\_k = r(x, y_k) - \frac{1}{K}\sum_{j=1}^K r(x, y_j)$
>
> **2. Inference output:**
>
> - BT outputs a single scalar value $r(x) \in \mathbb{R}$ through a scalar head on top of the base language model. This numerical score provides no insight into the evaluation reasoning process.
>
> - PaTaRM generates structured evaluation outputs consisting of: (1) dynamically generated  task-specific rubrics; (2) step-by-step reasoning process; (3) numerical ratings grounded in the generated rubrics and analysis.
>
> **3. Interpretability and Generalization:**
>
> - The nature of BT fundamentally limits it to modeling surface-level patterns in training data. As BT directly fits score differences from pairwise comparisons, it tends to overfit domain-specific features and lacks interpretability. Consequently, BT requires extensive domain-specific pairwise annotations for each new application scenario, leading to poor transferability.
>
> - By generating explicit evaluation criteria and reasoning, PaTaRM provides full transparency and stimulates the model's inherent evaluation capabilities rather than merely fitting data patterns. Our experiments demonstrate effective one-time training with robust multi-task transfer, achieving strong zero-shot performance on unseen instruction-following tasks (Table 2).
>
> **Summary:**
> BT trains a "scorer" that outputs opaque numerical values by fitting pairwise data patterns, while PaTaRM trains an "evaluation expert" that generates complete, interpretable reasoning traces.
>
>
> **Q2: The current framework could also suffer from the pairwise data bias.**
>
> A: Thank you for raising this concern. Our framework mitigates bias through three key mechanisms:
>
> **1. Bias mitigation mechanism:** PAR's relative-comparison objective (comparing each rollout score with the opposite group's average) is less sensitive to absolute label noise than point-wise regression, as preference flips are rarer than rating drifts.
>
> **2. Multi-rollout majority effect:** The multi-rollout design statistically stabilizes learning by averaging bias across multiple samples. This majority voting mechanism improves confidence and reduces noise impact on final rewards.
>
> **3. Data filtering strategies:** We implemented filtering strategies during data preparation to reduce inherent noise (detailed in the appendix), improving data quality and training reliability.
>
> While no method eliminates bias entirely, these mechanisms make PaTaRM more robust to label noise than standard BT or point-wise regressors.
>
> **Q3: It would be nice to see how the method compares when the labeled pairwise data is noisy.**
>
> A: Thank you for this valuable suggestion.
> - We conducted noise robustness experiments on Qwen3-8B, comparing BT and PaTaRM under varying label noise levels, as shown in the table. Additionally, we present detailed charts in the revised PDF.
>
> -  At 50% noise, BT collapses to 50.9% while PaTaRM maintains 81.3%, demonstrating superior robustness. At 100% noise, both models fail.
>
> - **Why PaTaRM is more robust:** BT directly fits all labels as supervision signals, forcing the model to learn from noisy data regardless of correctness. However, the PAR mechanism assigns rewards based on consistency with the model's inherent reasoning. Noisy labels conflicting with learned reasoning fail to provide effective reward signals, naturally filtering noise and maintaining stability.
>
> | FlippedRatio | 0% | 10%| 20%| 30%| 50%|
> |-|-|-|-|-|-|
> | BT-8B | 86.3 | 84.6 | 85.9 | 82.9 | 50.9 |
> | PaTaRM-8B | 84.2 | 84.2 | 84.7 | 82.9 | 81.3 |

---

> ### Author Response · Authors · 2025-11-23
>
> **Q4: It would be nice to see the performance on more reasoning heavy downstream tasks such as math and coding.**
>
> A: Thank you for the valuable suggestion. To validate our method on **reasoning-intensive tasks**, we conducted experiments on two representative mathematical reasoning benchmarks **(GSM-8K and Math-500)**. Due to time and computation constraints, we present results at the 96th training step—though not fully converged, this step sufficiently reflects performance trends of different reward mechanisms, as shown in the tables below.
>
> 1. **Training settings:** We used a merged dataset of GSM-8K and Math-500 training sets (11,973 samples) and maintained consistent inference configurations across all comparative experiments to ensure fairness.
>
> 2. **Overall performance:** PaTaRM exhibits significant effectiveness across model scales:
>
>    - Qwen3-8B (Strong Model): 5.8% relative improvement on Math-500 and 5.0% on GSM-8K compared to the base model;
>
>    - Qwen3-0.6B (Weak Model): 8.0% relative improvement on Math-500 and 5.2% on GSM-8K compared to the base model.
>
>
> 3. **Comparison with other reward mechanisms:** In Reinforcement Learning with Verifiable Rewards (RLVR) tasks, PaTaRM outperforms both rule-based reward models and Skywork-BT, with scale-specific characteristics:
>    - **Superiority over rule-based reward mechanism**: Rule-based rewards only provide sparse binary feedback (correct/incorrect) based on final answers, failing to capture intermediate reasoning rationality. **PaTaRM’s fine-grained process-oriented signals address this limitation**. For Qwen3-8B, PaTaRM hits 94.3% on GSM-8K (3.7% higher than rule-based) and 95.2% on Math-500 (slightly above rule-based 95.0%). For Qwen3-0.6B, PaTaRM still outperforms rule-based rewards on GSM-8K (81.0% vs. 78.4%) and on complex Math-500 (78.0% vs. 76.4%), showing process-aware advantages.
>    - **Superiority over Skywork-BT**: Skywork-BT **(generic BT) lacks pertinence to reasoning logic, leading to weak improvements.** **PaTaRM’s reasoning-specific design is more reliable.** Qwen3-8B with PaTaRM outperforms Skywork-BT on Math-500 (95.2% vs. 93.6%) and GSM-8K (94.3% vs. 93.7%). For Qwen3-0.6B, Skywork-BT barely improves (74.2% Math-500, 77.3% GSM-8K), while PaTaRM achieves 78.0% (8.0% relative gain) and 81.0% (5.2% relative gain), proving stronger adaptability.
>
>
> These results confirm **PaTaRM’s universality and effectiveness across scales**. It provides more informative guidance than **answer-only rewards** (e.g., rule-based) and more reliable signals than **generic BT rewards** (e.g., Skywork-BT).
>
> **Qwen3-8B Performance**
>
> | Method              | Math-500 |  GSM-8K  |
> | :------------------ | :------: | :------: |
> | Base Model          |   90.0   |   89.8   |
> | **+ PaTaRM**        | **95.2** | **94.3** |
> | + Rule-based Reward |   95.0   |   90.6   |
> | + Skywork-LLaMA-3.1-8B          |   93.6   |   93.7   |
>
> ---
>
> **Qwen3-0.6B Performance**
>
> | Method              | Math-500 |  GSM-8K  |
> | :------------------ | :------: | :------: |
> | Base Model          |   72.2   |   77.0   |
> | **+ PaTaRM**        | **78.0** | **81.0** |
> | + Rule-based Reward |   76.4   | 78.4 |
> | + Skywork-LLaMA-3.1-8B           |   74.2   |   77.3   |

---

> ### Author Response · Authors · 2025-11-26
> **Gentle reminder of the author-reviewer discussion deadline**
>
> Dear Reviewer 6o6v
>
> Thank you for your thoughtful and insightful comments! We sincerely appreciate your valuable feedback and suggestions. We are eager to engage in further discussions with you! In our previous responses, we have addressed all of the points you raised, supplementing the relevant experiments and clarifying any ambiguities. We believe that these revisions strengthen the contributions of our paper and look forward to your thoughts.
>
> As the discussion period deadline approaches, if you have any additional questions or concerns about the paper, we would be delighted to continue the conversation with you! We sincerely hope that our responses have effectively addressed your concerns and may encourage a more favorable reconsideration of our paper.
>
> Best regards

---

### Official Review · Reviewer_GYrz · 2025-11-01

**Soundness:** 2
**Presentation:** 3
**Contribution:** 3
**Rating:** 8
**Confidence:** 4

**Summary:**

This paper applies a language model post-training pipeline for generative reward models, allowing the model to generate the rubric and evaluate based on its rubric, namely PaTaRM. Notably, every component in post-training, including SFT, DPO, and RL training, is positively contributing to improving PaTaRM as a reward model. The empirical results on the most recent open-source models like Qwen3 demonstrate strong performance on both conventional reward modeling benchmarks and verifiable tasks like instruction-following.

**Strengths:**

1. The paper contributes to improving generative reward models, which is one of the emerging trends in reward modeling.
2. The experimental results on the Qwen3 series are promising, outperforming a few proprietary generative reward models.
3. Ablation study on SFT and RL training phases clearly supports that applying post-training for reward modeling is a valid approach for better human preference alignment.

**Weaknesses:**

The experimental design of the paper is reasonable, demonstrating the performance of PaTaRM as a reward model itself, and expanding its use in the RL training. However, one key experimental result is missing: Bradley-Terry reward model vs PaTaRM, both using Qwen3 as base models.

- **Comparison against the Qwen3 Bradley-Terry reward model**: While PaTaRM demonstrates promising results across the two reward model benchmarks, the choice of scalar reward models in Table 1 is questionable. Since the baseline performance of the Qwen3 series could be the dominant factor for the performance of PaTaRM, vanilla Bradley-Terry (BT) reward models trained on top of Qwen3-8B and 14B could build a stronger baseline for the claims in Section 4.2. For example, Skywork-Reward-V2 [1] series trained on Qwen3 show very strong performance on both RewardBench 2 and RMBench, while their training dataset is unknown. Given that, baselines trained with the same base models are essential to clearly state the empirical strength of the method.

&nbsp;

**References**

[1] Liu et al., 2025, “Skywork-Reward-V2: Scaling Preference Data Curation via Human-AI Synergy.” (Preprint)

**Questions:**

- There is a few literatures that claim Qwen models are exceptionally good at RL(VR) training. Would different model families, e.g., Gemma or Llama, benefit from the PaTaRM objective? Small-scale experiments like 3B scale would strengthen the effectiveness of the proposed method.

- How would the test-time scaling with thinking budget impact the final performance? Accuracy and thinking budget correlation analysis alongside the voting experiments in Section 4.7 would be interesting.

- Not a major limitation, but more samples generated from PaTaRM and the dataset for training PaTaRM in the Appendix will yield better understanding on the paper for the readers.

---

> ### Author Response · Authors · 2025-11-21
>
> ## **Title: Thank you for responses. We look forward to hearing back from you.**
>
> Dear Reviewer GYrz:
>
> Thank you for your responses and valuable questions. Below we provide our point-by-point responses.
>
> **W1:  Bradley-Terry (BT) reward models trained on top of Qwen3-8B and 14B could build a stronger baseline.**
>
> A: Thank you for the constructive feedback.
> - We conducted additional experiments to compare BT and PaTaRM models. We merged SFT and RL data (38.6k preference pairs) and trained BT models, as shown in the table below.
> - While BT models achieve higher scores on RewardBench, they exhibit significant weaknesses on RMBench, particularly on Hard subsets and Reasoning tasks. PaTaRM demonstrates superior performance on reasoning-intensive tasks.
> - These results validate that PaTaRM's generative reasoning paradigm provides **better robustness** on complex evaluation tasks, while BT models tend to **overfit superficial features** in data distribution. Based on BT's performance on RMBench, our training data may not be specifically optimized for improvement on this benchmark. However, PaTaRM can still enhance its reasoning abilities through the training process, leading to better performance on RMBench.
>
> | RewardBench | Chat | Chat Hard | Reasoning | Safety | Overall |
> |-|-|-|-|-|-|
> | BT-Qwen3-8B | 96.4 | 79.6 | 82.0 | 87.4 | 86.3 |
> | PaTaRM-Qwen3-8B| 91.0 | 71.5 | 87.9 | 86.3   | 84.2 |
> | BT-Qwen3-14B  | 95.3 | 87.5 | 89.2 | 87.6   | 89.9 |
> | PaTaRM-Qwen3-14B| 94.0| 73.9 | 91.7| 85.6 | 86.3 |
>
> |RMBench| Easy | Medium | Hard | Overall |
> |-|-|-|-|-|
> | BT-Qwen3-8B | 84.6 | 70.1| 56.2 | 70.3 |
> | PaTaRM-Qwen3-8B| 83.7 | 75.2| 64.6 | 74.5|
> | BT-Qwen3-14B | 85.8 | 70.7| 56.2 | 70.9 |
> | PaTaRM-Qwen3-14B| 86.0 | 76.9| 65.4 | 76.1|
>
>
> **Q1: Would different model families, e.g., Gemma or Llama, benefit from the PaTaRM objective**
>
> A: Thank you for the valuable suggestion regarding cross-model generalizability.
> - We conducted supplementary experiments using Llama-3.1-8B-Instruct as the base model to demonstrate that PaTaRM is not limited to the Qwen series and **generalizes effectively to other mainstream architectures**. As shown in the table.
> - The model achieved relative improvements of **6.8%** on RewardBench and **5.5%** on RMBench after RL training, validating the effectiveness of the PaTaRM framework across different model families.
> - **Tips**: We did not report the evaluation results of the initial model, because under the point evaluation template, the model struggles to generate output for answer extraction.
>
> | RewardBench| Chat | Chat Hard | Reasoning | Safety | Overall |
> |-|-|-|-|-|-|
> | Llama-3.1-8B-Instruct (SFT)   | 90.5 | 61.8| 63.1 | 79.3 | 73.7 |
> | Llama-3.1-8B-Instruct (RL)    | 90.5 | 69.7 | 71.2 | 82.7 | 78.7 |
>
> | RMBench | Easy | Medium | Hard | Overall |
> |-|-|-|-|-|
> | Llama-3.1-8B-Instruct (SFT)  | 71.2 | 62.9 | 52.6 | 62.2    |
> | Llama-3.1-8B-Instruct (RL) | 75.6 | 66.3 | 54.9 | 65.6    |
>
> **Q2: How would the test-time scaling with thinking budget impact the final performance?**
>
> A: Thank you for this insightful question. We conducted additional experiments to analyze the impact of inference budget on model performance as shown in the table below.
> - We referred to the settings in RM-R1[1]. For a fair comparison, we adjusted the max_token parameter to the corresponding inference budget during RL training, and applied the same length constraints during inference.
> - The results demonstrate that increasing the inference budget positively impacts overall performance.The most significant improvements occur in the Reasoning subset, which gets a **16.7%** gain. Chat Hard and Safety also show notable improvements of 6.4 and 5.1 points respectively. Interestingly, the Chat subset remains relatively stable, suggesting that standard conversational tasks do not require extensive reasoning budgets.
> - **Tips**:It should be noted that since our SFT data uses a token budget of 4096, this may influence the results, with the model performing optimally at its training budget.
>
> | Token | Chat | Chat Hard | Reasoning | Safety | Overall | Average Token |
> |-|-|-|-|-|-|-|
> | 1024  | 90.5 |65.1|71.2| 81.2| 77.0 | 310.6  |
> | 2048 | 89.7 | 64.7 |78.5| 82.3 | 78.8| 738.3 |
> | 4096 | 91.0 | 71.5| 87.9| 86.3| 84.2| 1500.8|
>
> [1] Xiusi Chen et al. RM-R1: Reward Modeling as Reasoning, 2025. https://arxiv.org/abs/2505.02387
>
> **Q3: Not a major limitation, but more samples generated from PaTaRM and the dataset for training PaTaRM in the Appendix will yield better understanding on the paper for the readers.**
>
> A: Thank you for this valuable suggestion. We completely agree that providing more concrete examples and detailed dataset information will enhance readers' understanding of our method. In the revised version, we have added the following content to the Appendix E.

---

### Author Response · Authors · 2025-11-28

Dear ICLR 2026 **AC, SAC, and PC**,

We would like to express our gratitude to all the reviewers for their valuable feedback. We have carefully considered all suggestions and updated our submission accordingly.

However, we have not yet received responses from **Reviewer GYrz, Reviewer 6o6v, and Reviewer YRyH**. With only few days remaining for discussion, we kindly request your assistance in reaching out to these reviewers. it would be greatly appreciated if you could encourage them to review our rebuttal, as we are eager to know if we have adequately addressed their questions and concerns.

We believe that constructive and timely communication between reviewers and authors is essential for the benefit of both parties.
Thank you for your hard work and support.

Best regards,

The authors of Paper Bridging Pairwise and Pointwise GRMs: Preference-Aware Reward Mechanism with Dynamic Rubric Adaptation

---

### Author Response · Authors · 2025-11-29
**Overall Response**

We sincerely thank the reviewers and the area chair for their insightful comments and constructive feedback. We are encouraged by the positive reception of our work and have carefully addressed each concern with additional experiments, clearer analysis, and targeted revisions to the manuscript.

### **Summary of Strengths Highlighted by Reviewers**

The reviewers have expressed strong interest in our proposed PaTaRM framework, with **Reviewer GYrz** giving a highly positive rating. The consensus on strengths includes:

*   **Innovative Framework Design:** **Reviewers Y2SM, 6o6v, and GYrz** acknowledged the method for effectively bridging the gap between pairwise preference data and pointwise supervision via the "PAR" mechanism, recognizing it as a valid approach for alignment.
*   **Interpretability via Dynamic Rubrics:** **Reviewer Y2SM** and **6o6v** highlighted the task-adaptive rubric generation as a key strength, providing fine-grained, interpretable evaluation criteria that **offer advantages over static methods**.
*   **Strong Empirical Performance:** **Reviewer GYrz** noted the promising results on Qwen3 (**comparing favorably to some proprietary models**) and confirmed that the ablation studies on the post-training pipeline (SFT, DPO, RL) clearly support the method's validity.
*   **Clarity and Modularity:** **Reviewers Y2SM** and **6o6v** noted that the paper is clearly written and presents a modular framework compatible with existing RLHF pipelines.

### **Core Contributions of Our Work**

1.  **Unified Preference-Aware Framework:** We propose **PaTaRM**, integrating a **Preference-Aware Reward (PAR)** mechanism with dynamic rubric adaptation. PAR leverages pairwise signals to capture quality gaps, **addressing the dependency on explicit pointwise labels**.
2.  **Dynamic Rubric Adaptation:** We introduce a mechanism to generate task-level and instance-specific criteria. This enables GRMs **to** assess responses with high granularity, **mitigating** the adaptability limitations of static rubrics.
3.  **Empirical & Downstream Impact:** PaTaRM achieves a **5.5%** relative improvement on RewardBench and RMBench. Crucially, in downstream RLHF, it delivers a **13.6%** gain on IFEval and InFoBench, **performing favorably compared to baselines in OOD tasks**.
4.  **Inference Efficiency:** We clarify that PaTaRM operates with **$O(n)$** inference complexity, offering a scalable alternative to Pairwise Generative RMs ($O(n \log n)$) while providing richer signals than scalar RMs.

### **Addressing Main Concerns**

We have extensively addressed the reviewers' concerns through rigorous new experiments and theoretical clarifications:

**1. Baseline Comparisons: PaTaRM vs. Bradley-Terry RM**
*   **Concern:** Reviewers requested a direct comparison with BT model.
*   **Response:** We added the requested BT baseline. Results show **PaTaRM demonstrates notable gains over BT on reasoning-heavy tasks** (e.g., RMBench Hard) and demonstrates **enhanced** robustness to label noise, confirming it learns evaluation logic rather than merely fitting score differences.

**2. Computational Cost and Complexity Definition**
*   **Concern:** Questions regarding training latency and complexity gaps.
*   **Response:**
    *   **Cost:** We reported a ~25-39% training overhead, which we argue is a worthy trade-off for the interpretability and OOD generalization gains.
    *   **Complexity:** We clarified that PaTaRM functions as a **Pointwise Generative RM** with **$O(n)$** inference complexity, offering **improved** efficiency compared to the $O(n \log n)$ required by Pairwise Generative RMs.

**3. Generalization to Other Architectures and Tasks**
*   **Concern:** Applicability to different model families and downstream reasoning tasks.
*   **Response:** We validated cross-model effectiveness on **Llama-3.1-8B** (+5.5% on RMBench) and extended testing to reasoning tasks (**GSM-8K, Math-500**), where PaTaRM consistently **achieved higher scores than** Rule-based and Skywork-BT baselines.

**4. Reward Hacking and Global Consistency**
*   **Concern:** Risks of "self-optimization" and lack of global transitivity.
*   **Response:**
    *   **Consistency:** We explained that our optimization objective implicitly **promotes transitivity, empirically validated by stable rollout rankings**.
    *   **Hacking:** We detailed the "Primary Rubric" constraint mechanism to anchor generation; the **robust performance on unseen tasks** further evidences our method's resilience against reward hacking.

We sincerely thank the reviewers and the Area Chair again for their thoughtful comments, which have helped us significantly strengthen the paper with the new experiments, refined analysis, and clarifications above. We believe our work provides a practically useful contribution to the field by establishing a robust framework that effectively bridges pairwise preference data with interpretable, reasoning-based pointwise reward modeling for RLHF.

---

### Note · Authors · 2026-01-04

I have read and agree with the venue's withdrawal policy on behalf of myself and my co-authors.